# STARFIRE: REGULARIZATION FREE ADVERSARIALLY ROBUST STRUCTURED SPARSE TRAINING

## ABSTRACT

This paper studies structured sparse training of CNNs with a gradual pruning technique that leads to fixed, sparse weight matrices after a set number of epochs. We simplify the structure of the enforced sparsity so that it reduces overhead caused by regularization. The proposed training methodology explores several options for structured sparsity.

We study various tradeoffs with respect to pruning duration, learning-rate configuration, and the total length of training. We show that our method creates a sparse version of ResNet50 and ResNet50v1.5 on full ImageNet while remaining within a negligible <1% margin of accuracy loss. To ensure that this type of sparse training does not harm the robustness of the network, we also demonstrate how the network behaves in the presence of adversarial attacks. Our results show that with 70% target sparsity, over 75% top-1 accuracy is achievable.

## 1    INTRODUCTION

Pruning weights can compress a network into a smaller model so that the model can fit into faster/smaller memory and therefore result in execution speedups (Han et al., 2016; 2015a). To increase the accuracy Han et al. (2015b) and Mao et al. (2017) explore training the network dense after pruning. The resulting network can maintain accuracy based on the specified level of sparsity (Mostafa & Wang, 2019; Zhu & Gupta, 2017; Han et al., 2015a).

Structured sparsity has been explored for RNNs and also CNNs where a certain number of non-zeros is allowed across various cross-sections of the weight tensors. These methods aim to speed up computation and reach some final level of sparsity for deployment. Narang et al. (2017) have shown promising results for structured training of RNNs while sparse CNNs could not achieve the same performance (Mao et al., 2017).

Recent work has demonstrated that structurally sparse training can speed up execution on GPUs (He et al., 2017; Lym et al., 2019; Zhu & Gupta, 2017). However, these training mechanisms add regularization and computational overhead to eliminate unnecessary weights. The regularization term modifies the original training and can be expensive in hardware. While enforcing coarse-grain sparsity Lym et al. (2019) provides significant speedups, the final network contains an insufficient degree of sparsity for deployment on edge devices.

Mostafa & Wang (2019) show that with adaptive sparse training and dynamic reallocation of non-zeros sparsity levels up to 80% can be achieved. However, even though an additional 10 epochs of training are required, an accuracy loss of around 1-2% is still observed. The main drawback is the overhead incurred while implementing such technique on the target platform. Continuous reconfiguration of the sparsity pattern is expensive as it does not allow for compression of weights during training.

To achieve speedups and a desired final degree of sparsity, we aim to apply the techniques in Han et al. (2015b) and Mao et al. (2017) at earlier stages in training at higher frequency within a period which we call the pruning era, usually a period of 20-30 epochs. During the pruning era, with fine granularity of at most a kernel size, we exploit one of the three proposed sparsity regimes. Subsequently, we fix the mask for the rest of the training to speed it up. Our motivation came from the insight that having a *fixed* sparse multiply-accumulate pattern allows weight compression during training and can save compute and energy in hardware (Han et al., 2016).

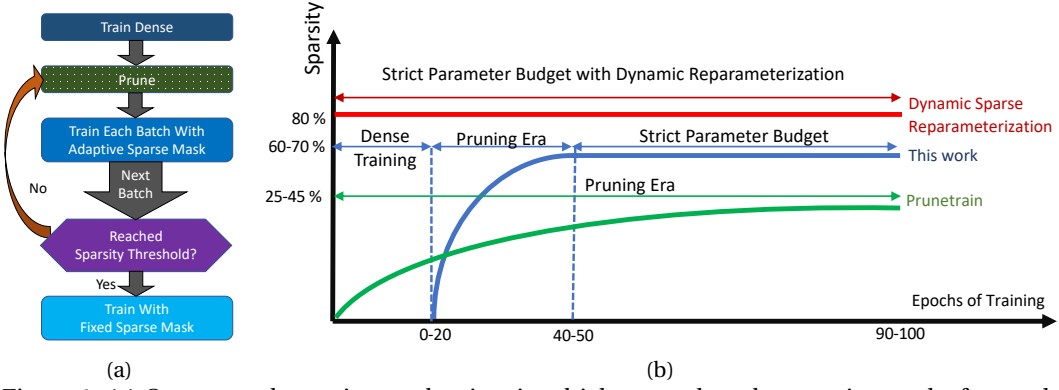

(a)                                              (b)

Figure 1: (a) Our general sparsity mechanism in which we update the sparsity mask after each batch until we reach a desired level of sparsity. (b) Schedule comparison between this work, Mostafa & Wang (2019), and Lym et al. (2019). Our work has the shortest pruning era and more gradually reaches our final sparsity.

We explore the impact of various pruning granularities, sparsity levels, and learning-rate schedules on the network's convergence as well as adversarial robustness for CNNs like Resnet-50 (He et al., 2015) on ImageNet and tinyImagenet (CS231N, 2015).

Recent literature has shown that adversarial attacks are more successful on pruned neural networks than they are on regular neural networks (Wang et al., 2018). Given the danger of adversarial attacks in real world situations, we find that it is important to evaluate our sparsity techniques under adversarial robustness. We leverage the FGSM mechanism (Goodfellow et al., 2014) to evaluate the adversarial robustness on our sparse models. This paper makes the following contributions:

1. We propose a mechanism to train and prune a convolutional network during the earlier stages of training such that this sparsity can be harvested for the computational speedups. To do this, we fix the sparse weight masks for the remainder of the training.

2. For fully connected sparsification, we eliminate blocks of fully connected weights based on their connection to the zeros in the previous convolutional layer.

3. We enforce structural, regularization free, magnitude-based pruning across two distinct dimensions and a combined version. These dimensions are inside convolution window $R \times S$ and across input/output feature matrix ($CK$).

4. Our sparse models are as robust to adversarial FGSM attacks as fully dense models.

5. We demonstrate that early stage dense training is crucial for maintaining high accuracy.

6. The proposed technique is tolerant to sparsity levels of up to 60-70% with under 1% accuracy degradation. We can compensate by scheduling an extra learning rate drop and training for an extra 10 epochs.

The rest of the paper is organized as follows. Section 2 explains our pruning methodology. Section 3 describes the experimental setup framework. Section 4 presents results and discusses their interpretation. Section 5 presents the related work. Section 6 concludes the paper.

## 2 PRUNING METHODOLOGY

Our proposed pruning mechanism works by always pruning the weights of smallest magnitude after each weight update. After a forward and backward pass (one batch update), the model is pruned. If a weight is already zero, the gradient is also set to zero. This means that once a weight becomes zero, it will remain zero for the rest of the training period.

This mechanism is similar to Han et al. (2015b), except that we only prune in the earlier stages of the training as opposed to post training. Additionally, this work is similar to Narang et al. (2017) although we set the sparsity threshold instead of using a heuristic to calculate it. We chose this pruning mechanism because of its negligible computational overhead.

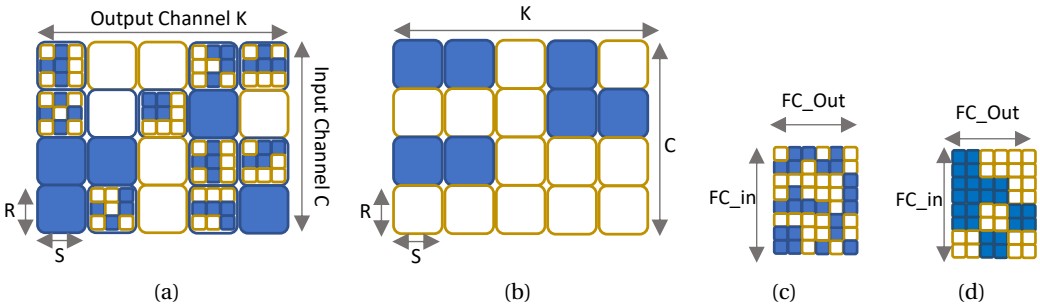

Figure 2: This figure illustrates sparsity across different weight dimensions and different granularities. (a) *Window sparsity*, which prunes the 5 smallest weights in a 3×3 window. (b)*CK pruning* where whole R×S convolutional kernels are pruned. (c) Sparsity in fully-connected weights. (d) Block sparsity in a fully-connected layer.

In our pruning algorithm, the sparsity threshold refers to the percentage of weights in the network that are currently pruned. Before or during the first epoch of pruning, we will have a sparsity threshold of zero. As we continue training, we gradually increase the sparsity threshold so that by the final epoch of pruning the network sparsity will have reached our final, desired threshold. Finally, we also define the *pruning era* to be the epochs between the first and final epochs of pruning depicted in Figure 1b.

We evaluate the pruning mask after every training step until we reach the final epoch of pruning. After the final epoch, the pruned values in the network will remain zero for the rest of training; no new pruning will occur, and only the non-zero weights will be updated.

## 2.1 PRUNING METHODOLOGY BY LAYER

Pruning the smallest magnitude weights in the *entire* network is inefficient because it involves sorting the weights over the network. Instead, we prune the smallest magnitude weights or sum of weights, within a certain locale of the network. When pruning, we examine each layer individually and apply a separate technique to evaluate which weights to prune, depending on the type of layer we are currently pruning.

### 2.1.1 CONVOLUTIONAL LAYER PRUNING

**Window pruning for 3x3 Convolutional Layers** Figure 2a shows the result of a pruned 3×3 convolutional weight tensor under the window pruning method. In this scheme, window layer pruning refers to pruning of weights within the 3×3 convolution kernels. We allow a maximum number of non-zero values for each kernel in the 3×3 convolutional layers and eliminate the weights of smallest magnitude.

---

**Algorithm 1** CK Pruning Algorithm

generate_ck_sparsity_mask($\theta_{layer}$, sparsity_threshold):
**for** $\theta$ in $\theta_{layer}$ **do**
   **for** all c in C **do**
      **for** all k in K **do**
         kernel_max$_{c,k}$ = max($\theta_{c,k}$)
      **end for**
      cutoff_index = size($\theta_c$) * sparsity_threshold
      n = max(cutoff_index, size($\theta_c$) − max_non_zero − 1)
      cutoff_value = $n^{th}$ largest value in kernel_max$_c$
      **for** all k in K **do**
         mask$_{c,k}$ = 1 if kernel_max$_{c,k}$ > cutoff_value, else 0
      **end for**
   **end for**
**end for**

---

**CK Pruning Methodology** Figure 2b shows the result of a pruned 3×3 convolutional weight tensor under the CK pruning method. In this scheme, the weights of a certain layer can be viewed

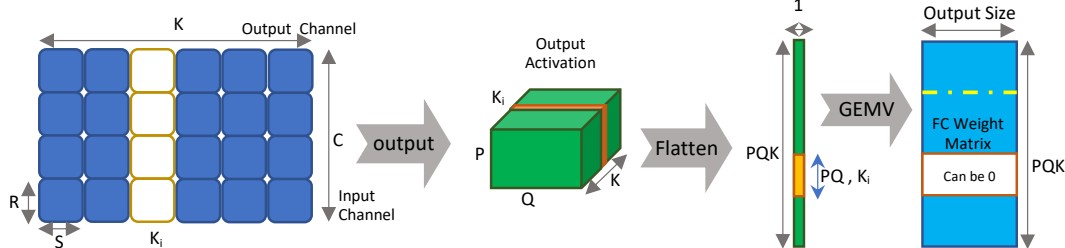

Figure 3: FC pruning taking advantage of the sparsity in the previous convolution layer.

as a CK matrix of R×S kernels. The CK pruning method involves pruning the 3×3 convolutions along the channel and kernel dimensions of each convolutional filter, i.e., we prune whole kernels (CK matrix of R×S windows) at once and can ultimately prune all the input channels in an output channel. As defined by Algorithm 1, we determine which filter to prune by examining the max of the magnitudes of all the weights in a kernel, which is the max of nine weights. This max is used to evaluate whether the whole kernel should be pruned or not.

**Combined Pruning Methodology**   To combine window and CK pruning, we introduce an *intra-epoch combined pruning method*, which we refer to hereafter as "intra-epoch pruning" or "intra", for short. As shown by appendix Algorithm 4 in the Appendix, in a given epoch we first apply window pruning to each 3×3 convolutional layer at a fraction of the *sparsity threshold* for that epoch. Then, we prune the remaining fraction of the *sparsity threshold* with CK Pruning.

### 2.1.2   FULLY CONNECTED PRUNING

Like pruning for convolutional layers, we apply a two-tier pruning scheme from Mao et al. (2017) for fully connected layers: micro-level pruning within a block and macro-level pruning that eliminates entire blocks.

**Block FC Pruning**   Figure 2d refers to pruning of individual blocks. Here, we prune an entire n×n (n<5) window within the dense layer and create coarse grained sparsity. To do this, we sum the magnitude of the weights in each window and prune the windows with the smallest magnitude.

**Fine FC Pruning**   Figure 2c refers to the pruning of individual weights. Here, we prune the individual weights in the entire FC Layer, where we compare the magnitude of all the weights to each other.

The produced zero patterns in the last convolution layer allow for eliminating more weights in fully connected layer as depicted in Figure 3. If all the $C$ windows for a specific $K_i$ are zeros, the output activation for the corresponding $K_i$ is also zero. The corresponding neurons in the following fully connected layer are therefore receiving zero input activations and can be eliminated along with their associated weights. This enables us to get sparsity without having to evaluate the weights in the fully connected layer.

When pruning just the small weights in the FC layer, one can inadvertently cut off relevant connections between the input and output layers. Accordingly, we structure the pruning mechanism such that each output neuron should be influenced by the input. This means every column in the weight matrix of the fully connected layer in Figure 3 has at least one non-zero element.

## 3   EXPERIMENTAL SETUP

To validate each type of pruning (window, CK, or intra-epoch) we selected ResNet50 (He et al., 2015) v1 and v1.5 with the ImageNet and/or Tiny-ImageNet (CS231N, 2015) datasets. We evaluated each pruning method by varying *sparsity levels* and *pruning era*.

We experimented with ResNet50 v1.5, in addition to v1, to explore how changing the network structure would affect the top-1 accuracy. For window pruning, we tested with ResNet50v1 on Tiny-ImageNet as well as ResNet50v1 and v1.5 on ImageNet to compare the impact of strided convolutions on our sparse training. Also, we experimented with the learning rate schedule of the training regime. Our typical schedule for ResNet50v1.5 included learning rate drops at epochs 30,

60, and 90, but we experimented with placing the last drop at epoch 80 instead. Unlike typical ResNet50 training, which uses a batch size of 256 and starts the learning rate at 0.1, we used batch size 64 as this is what could fit in our GPUs. As suggested by Krizhevsky (2014), we scaled the starting learning rate by $\frac{1}{\sqrt{4}} = \frac{1}{2}$ to 0.05 in order to compensate for the smaller batch size.

### 3.1 SPARSE TRAINING EXPERIMENTS

For ResNet50v1 and Tiny-ImageNet, we did gradual pruning until epoch 10. We subsequently enforced the final sparsity requirement, set a maximum number of non-zero values in each window/kernel of each layer, and fixed this sparsity pattern for the rest of training. We chose the 10th epoch as the *final epoch of pruning* because we wanted to see if we could fix the sparsity mask early in the training process.

For ResNet50v1 and ImageNet, our goal was to start pruning as early as possible, while maintaining high accuracy. We set our *pruning era* to epochs 0-30. Our hypothesis was that 30th epoch would be a suitable epoch to stop pruning, because this is where the learning rate is first decreased, in addition, there would be a large drop in accuracy if we stop pruning at epoch 20. However, this schedule did not perform well for ResNet50v1.5 and ImageNet and therefore we set our *pruning era* to 30-50.

To test training using CK and intra-epoch pruning, we mainly used ResNet50v1 and ResNet50v1.5 with ImageNet, but also performed CK pruning on ResNet50v1 and Tiny-ImageNet. We adopted a similar approach to Han et al. (2015b) to train with CK or intra-epoch pruning by setting the *first epoch of pruning* to 20, 30, or 40 with a *pruning era* of 20 or 30 epochs. Then, we continued to train the sparsified network until the final epoch.

### 3.2 ADVERSARIAL ROBUSTNESS

Since there was evidence that increasing sparsity lowers adversarial robustness (Wang et al., 2018), we evaluated this robustness in our models. To do so, we applied *Fast Gradient Sign Method* (FGSM) attacks defined in Goodfellow et al. (2014) on one of our sparse models, to generate its own adversarial examples, and measured the validation accuracy again. We used the same validation set as ImageNet and applied the attack's image transformation to each input image. Moreover, we experimented with a variety of different $\epsilon$ in order to see how our accuracy decayed. Lastly, in our experiments we leveraged the examples provided in Pytorch tutorials [1].

## 4 RESULTS

### 4.1 RESNET50 ON TINY-IMAGENET

From our experiments with Tiny-Imagenet (shown in Appendix in Table 5), we see that even with up to 80% sparsity, both window and CK pruning are able to achieve levels of accuracy comparable to the dense baseline. CK pruning performs even better than the baseline.

### 4.2 RESNET50 ON IMAGENET

Our ResNet50 v1.5 experiments (Table 1 and Appendix Figure 11) with the *first epoch of pruning* at epoch 30 show that all of our pruning methods are able to achieve over 73% accuracy, and we can achieve above 74% accuracy up to 70% sparsity. Table 2 shows that on ResNet50 v1, our methods can achieve between 0.1-0.3% less than the baseline.

By comparing the sparsity curves of the window, CK, and intra-epoch pruning runs in Figure 4 (top right), we observe that the sparsity of window pruning is not as smooth as the other methods. This is likely indicative of the more rigid structure of CK and intra-epoch pruning, which causes the degree of sparsity to be much more uniform from epoch to epoch.

Figure 4 (top left, bottom right) also shows on ResNetv1.5, the window is slightly better than the CK and intra, which have similar performance, but the window is worse than the other two on

---

[1] https://pytorch.org/tutorials/beginner/fgsm_tutorial.html

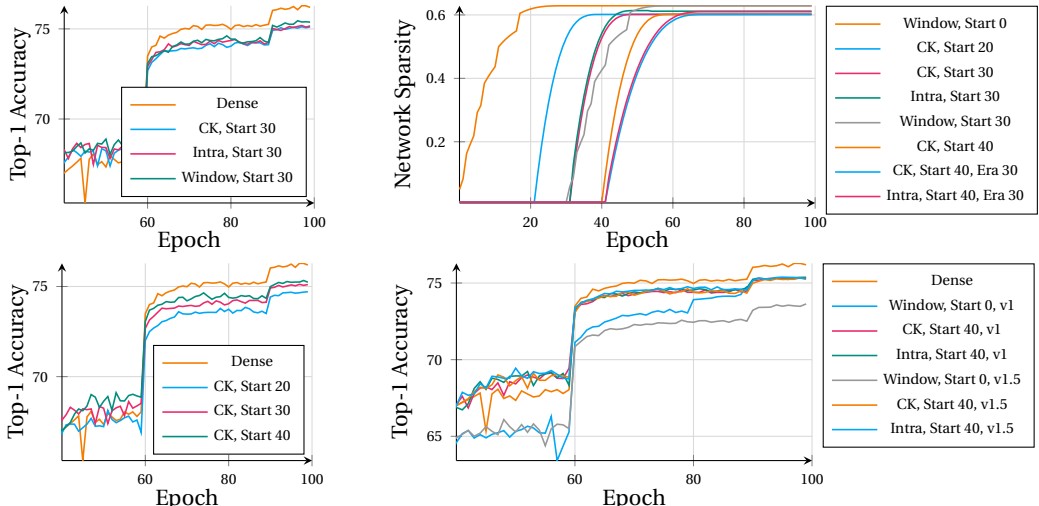

Figure 4: top left: Convergence plots of all pruning methods with first epoch of pruning 30, top right: Sparsity plot of all methods, bottom left: and CK starting pruning at different epochs, bottom right: Resnet v1 and v1.5 at 60% sparsity for 90 epochs.

| Model Sparsity (%) | 40 | 60 | 70 | 80 |
|---|---|---|---|---|
| Dense (76.29) | - | - | - | - |
| CK, start 40 | 75.82 (-0.46) | 75.33 (-0.95) | 74.92 (-1.36) | 74.16 (-2.12) |
| CK, start 30 | 75.84 (-0.45) | 75.12 (-1.17) | 74.72 (-1.56) | 73.66 (-2.63) |
| CK, start 20 | 75.55 (-0.74) | 74.71 (-1.58) | 74.32 (-1.96) | 72.79 (-3.50) |
| Intra, start 40 | 75.89 (-0.39) | 75.38 (-0.90) | 75.07 (-1.21) | 74.02 (-2.27) |
| Intra, start 30 | 75.84 (-0.45) | 75.16 (-1.12) | 74.48 (-1.80) | 73.66 (-2.63) |
| Intra, start 20 | 75.72 (-0.57) | 74.75 (-1.53) | 74.26 (-2.02) | 72.97 (-3.32) |
| Window, start 0 | - | 73.63 (-2.65) | 72.79 (-3.50) | 70.25 (-6.04) |
| Window, start 30 | - | 75.45 (-0.84) | 74.65 (-1.63) | 73.31 (-2.98) |
| CK, start 40, era 40-70 | - | 75.52 (-0.77) | 75.16 (-1.13) | - |
| Intra, start 40, era 40-70 | - | 75.56 (-0.73) | 75.14 (-1.15) | - |

Table 1: Main results, top-1 accuracy on Resnet50 v1.5 after 100 epochs.

ResNetv1. Furthermore, starting the *pruning era* later improves performance (Figure 4-(bottom left)).

| Model | Epoch 90 | | Epoch 100 | |
|---|---|---|---|---|
| | Sparsity [%] | Accuracy [%] | Sparsity [%] | Accuracy [%] |
| CK, start 40 | 60 | 74.60 | 60 | 75.36 |
| Intra, start 40 | 60 | 74.68 | 60 | 75.37 |
| Window, start 0 | 60 | 74.78 | - | - |
| CK, start 30 | - | - | 80 | 73.66 |
| PruneTrain ((Lym et al., 2019)) | 50 | 73.0 | - | - |
| Dyn Sparse ((Mostafa & Wang, 2019)) | - | - | 80 | 73.3 |
| Dyn Sparse (kernel granularity) | - | - | 80 | 72.6 |

Table 2: Main results, top-1 accuracy on ResNet50 v1 after 90 and 100 epochs. Comparison with related work. PruneTrain sparsity is not explicitly stated, so we estimate their sparsity level from their inference FLOPs saved (1-FLOPs). Also, our experiments were run with batch size 64.

Table 3 demonstrates that our sparsity mechanism can have a minimal drop in adversarial robustness (approximately 1-1.5%) compared to the dense baseline model, whereas other methods see more accuracy degradation (Wang et al., 2018).

The sparsity of each layer, depicted in Figure 5, emphasizes that early layers tolerate sparsity better, as they have consistently higher sparsity in the last 1×1 convolutional layer of each residual block. This may be due to their vicinity to the residual connection, which provides additional information to the layer.

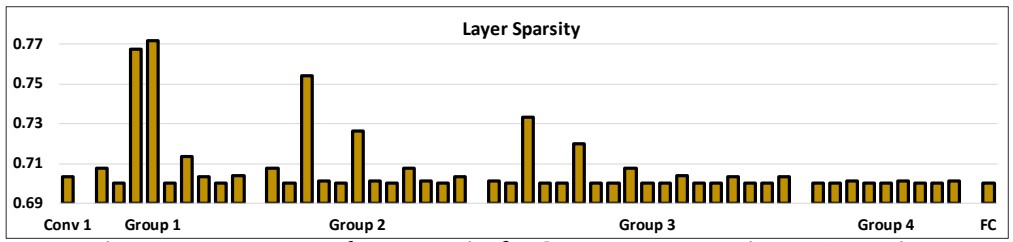

Figure 5: Resnet50 v1.5 layer sparsity for CK, start 30, targetting 70% sparsity.

| Model | Sparsity | $\epsilon = .05$ | $\epsilon = .1$ | $\epsilon = .15$ | $\epsilon = .2$ | $\epsilon = .25$ | $\epsilon = .3$ |
|---|---|---|---|---|---|---|---|
| Dense | 0 | 41.71 | 30.47 | 25.24 | 22.39 | 20.69 | 19.58 |
| Intra, start 40 | 0.6 | 40.42 | 29.02 | 23.67 | 20.73 | 18.93 | 17.73 |
| Intra, start 40 | 0.7 | 40.03 | 28.55 | 23.13 | 20.13 | 18.32 | 17.10 |
| Window, start 30 | 0.6 | 40.52 | 29.12 | 23.77 | 20.81 | 18.99 | 17.78 |
| Window, start 30 | 0.7 | 39.73 | 28.32 | 22.91 | 19.88 | 18.02 | 16.77 |
| CK, start 40 | 0.6 | 40.51 | 29.21 | 23.95 | 21.08 | 19.34 | 18.18 |
| CK, start 30 | 0.7 | 39.83 | 28.44 | 23.09 | 20.11 | 18.31 | 17.12 |

Table 3: Adversarial Robustness of Resnet50 v1.5 on Imagenet

## 4.3 DISCUSSION

Overall, we notice that there is a tolerance for sparsity (up to 70%), which yields around 1% accuracy loss compared to the dense baseline. However, this loss can be compensated by dropping the learning rate and performing another 10 epochs of training, which provides a 0.7-0.9% accuracy increase. With high levels of sparsity this extension is computationally cheap.

We observed the early stages of dense training are important for high accuracy, as longer periods of dense training consistently outperformed shorter ones. Moreover, widening the pruning era slightly (10 epochs) improves the final convergence accuracy (by around 0.2%).

We also observed that pushing the learning rate drop schedule to earlier epochs or aligning it with pruning era does not improve the final accuracy. However, pushing the last learning rate drop from epoch 90 to 80 can improve the accuracy by around 0.1%. (See Appendix Table 8 and Table 1)

We postulate that window pruning performs worse for ResNetv1.5 compared to ResNetv1 due to the strided nature of convolutions in ResNetv1.5.

## 5 RELATED WORK

To give a broad comparison stage, we extended Mostafa & Wang (2019)'s table on alternative sparsity mechanisms in Table 4 with respect to characteristics of their mechanisms: training/compression focus, regularization, the period in which pruning is applied, strictness of parameter budget, and pruning granularity. We explain each of the columns below:

1. **Training Focus**: Trying to train while maintaining/increasing sparsity of the network. The opposite is **Compression Focus**, i.e., methods that only seek to provide a smaller network for inference.
2. **Regularization**: Applying a regularization value to the loss, in order to find and prune irrelevant weights, while others use magnitude-based pruning.
3. **Pruning Era**: The period during training in which the pruning is applied.
4. **Strictness of Parameter Budget Era wrt to Pruning**: A strict parameter budget is fixed to the size of the final sparse model. Mostafa & Wang (2019) have a strict budget throughout training. Our method is only strict after the pruning era. Some networks do not have a strict parameter budget and only prune weights that appear to be irrelevant and without a sparsity target.
5. **Pruning Granularity**: The level of granularity within in the network at which values are pruned. For example, at the kernel level, we determine which values to prune by examining only the weights in the kernel (Mao et al., 2017). See Figure 2 for more information.

We chose these concepts because their characteristics can enable faster and lower-energy training. A strict parameter allows the hardware mapping to plan for a fixed number of multiply-accumulate

| Method | Train/ Cmprss Focus | Requires Regular -ization | Pruning Era | Strict Parameter Budget Era | Granularity of Sparsity |
|---|---|---|---|---|---|
| Window (This Work) | T | No | Beginning | After Pruning | non in Window |
| CK/Intra-Epoch (This Work) | T | No | Middle | After Pruning | Kernel |
| Evolutionary (Mocanu et al., 2018) | T | No | Beginning | Throughout | non-structured |
| (Zhu & Gupta, 2017) | T | No | Throughout | After Pruning | non-structured |
| Lottery (Frankle & Carbin, 2018) | T | No | Throughout | Throughout | non-structured |
| RNN Pruning (Narang et al., 2017) | T | No | Beginning | None | non-structured |
| NeST(Dai et al., 2017) | T | No | Throughout | None | non-structured |
| PruneTrain (Lym et al., 2019) | T | Yes | Throughout | After Pruning | Layer/Channel |
| Dyn Sparse (Mostafa & Wang, 2019) | T | Yes | Throughout | Throughout | non-/Kernel |
| DeepR (Bellec et al., 2017) | T | Yes | Throughout | Throughout | non-structured |
| Deep Comp (Han et al., 2015b) | C | No | Throughout | - | non-structured |
| L1-Norm Channel (Li et al., 2016) | C | Yes | Throughout | - | Channel |
| (Lebedev & Lempitsky, 2016) | C | Yes | End | - | non-structured |
| Sparsity Gran (Mao et al., 2017) | C | Yes | Throughout | - | non-structured |
| SSL (Wen et al., 2016) | C | Yes | Throughout | - | Channel/Kernel/Layer |
| ThiNet (Luo et al., 2017) | C | Yes | End | - | Channel |
| LASSO-regression (He et al., 2017) | C | Yes | End | - | Channel |
| Slimming (Liu et al., 2017) | C | Yes | Throughout | - | Channel |
| SSS (Huang & Wang, 2018) | C | Yes | Throughout | - | Layer |
| PFA (Suau et al., 2018) | C | Yes | Throughout | - | Channel |

Table 4: Comparison of Training Methods that yield sparse networks

operations. The granularity of the sparsity mechanism indicates how easy it is to adapt the mechanism to an existing hardware. The coarser the granularity, the more adaptable it is to existing hardware (Mao et al., 2017). Regularization, although useful in forcing the network to learn prunable weights, adds more irregularity to computation flow. The pruning era starting at the beginning of the training enables us to train with a compressed network.

Mao et al. (2017) explores pruning on a range of granularities including window, kernel, and filter, and their effect on accuracy. They also qualitatively and quantitatively show that coarse-grain pruning, like kernel- or filter-level sparsity, is more energy-efficient due to fewer memory references. Similarly, our work surveys sparsity at the window, kernel, and filter levels. We improve on Mao et al.'s work in two ways. First, we show higher top-5 accuracy at higher sparsity levels on a complex benchmark, ImageNet on ResNet50 (92.338% at 40% CK sparsity), and we also show high top-1 accuracy whereas Mao et al. only report top-5.

Prunetrain (Lym et al., 2019) explores a way to create sparse channels and even layers to speed up training with around a 1% drop in accuracy. However, this requires a shift in the training mechanism, including a regularization term that could effect how the mechanism scales to large and distributed settings and that must be computed throughout training. The resulting network is only around 50% sparse and the accuracy loss due to sparse training is high enough that a baseline network with same accuracy could result into same computational savings by just terminating training at much earlier stage/epoch.

In contrast to other pruning mechanisms, our proposed window, CK, and combined sparsity mechanisms have strict parameter budgets after the pruning era. The CK and combined schemes have channel-level and kernel-level pruning granularities.

## 6    CONCLUSION AND FUTURE WORK

In this work, we introduced techniques to train CNNs with structured sparsity and studied the tradeoffs associated with various implementation options. We demonstrated on ResNet50 with the full ImageNet dataset that the proposed sparse training method outperforms all related work and is comparable to a dense model in terms of convergence accuracy. We also observed that delaying the start of enforced, gradual pruning to at least epoch 20 was necessary to reach high convergence accuracy, highlighting the importance of the early epochs of dense training. Moreover, performing an additional 10 epochs of training provides substantial (around 1%) accuracy gains of the final model. In the future, we would like to study the tradeoffs of sparse training on low-precision networks.

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

## 7 APPENDIX

### 7.1 DETAILS OF PRUNING ALGORITHMS

Here we provide full descriptions of our other pruning methods and our general methodology sparse training.

**Sparse Training Methodology**   Algorithm 2 shows how we modify normal training in order to train sparsely.

---
**Algorithm 2** Pruning Algorithm
---
current_iter = 0
**while** training **do**
  **if** current_iter > *first epoch of pruning* and current_iter < *last epoch of pruning* **then**
    *mask* = generate_sparsity_mask( $\boldsymbol{\theta}$, current_iter, *sparsity threshold* )
  **end if**
  $\boldsymbol{\theta}_{pruned} = mask \cap \boldsymbol{\theta}$
  $\hat{\boldsymbol{y}}$ = forward_pass( $\boldsymbol{\theta}_{pruned}$, $\boldsymbol{x}$ )
  $\boldsymbol{\theta}$ = weight_update( $\boldsymbol{y}$, $\hat{\boldsymbol{y}}$, $\boldsymbol{\theta}_{pruned}$)
  current_iter = current_iter + 1
**end while**

---

**Window Pruning Methodology**   Algorithm 3 shows how we prune with window sparsity.

---
**Algorithm 3** Window Pruning Algorithm
---
generate_window_sparsity_mask($\theta_{layer}$, sparsity_threshold):
**for** $\theta$ in $\theta_{layer}$ **do**
  **for** all c in C **do**
    **for** all k in K **do**
      cutoff_index = size($\theta_{c,k}$) $*$ sparsity_threshold
      n = max(cutoff_index, size($\theta_{c,k}$) $-$ max_non_zero $- 1$)
      cutoff_value = $n^{th}$ largest value in $\theta_{c,k}$
      **for** all i,j in R,S **do**
        $mask_{i,j,c,k} = 1$ if $\theta_{i,j,c,k} >$ cutoff_value, else 0
      **end for**
    **end for**
  **end for**
**end for**

---

**Combined Pruning Methodology**   To combine Window and CK pruning, we introduce *intra-epoch pruning*. As shown by Algorithm 4, in a given epoch we first apply Window Pruning to each 3×3 convolutional layer at a fraction of the *sparsity threshold* for that epoch. Then, we prune the remaining fraction of the *sparsity threshold* with CK Pruning. The idea being that kernels that lose many of their parameters during window pruning can be fully pruned during the CK pruning phase. Our intuition is that first pruning parameters within a kernels guides the subsequent CK pruning towards the less important kernels. Thus, we pick out better kernels to prune. We also gain more structured sparsity but sacrifice the precision of window pruning.

---
**Algorithm 4** Intra-Epoch Pruning Algorithm
---
generate_intra_epoch_sparsity_mask($\theta_{layer}$, sparsity_threshold):
**for** $\theta$ in $\theta_{layer}$ **do**
  window_mask = generate_ck_sparsity_mask($\theta$, sparsity_threshold)
  ck_mask = generate_ck_sparsity_mask($\theta$, sparsity_threshold)
  mask = window_mask **and** ck_mask
**end for**

---

For completeness, we also tried another method of combining called *inter-epoch pruning*, which involved splitting the *pruning era* into CK pruning and window pruning phases. However, from our initial experiments we determined that intra-epoch pruning, performed better (though it was more computationally expensive) than inter-epoch pruning. With inter-epoch pruning we were only able to achieve 74.1% top-1 accuracy with a *first epoch of sparsity* of 40 and a *final sparsity* of 40% on Resnet50v1.5 and Imagnet. The same setup trained with intra-epoch pruning achieved 74.9% accuracy. Thus, we pursued intra-epoch pruning as our method to combine the two sparsification methods.

### 7.2 ADDITIONAL DETAILS ON EXPERIMENTAL SETUP

This section goes into more detail on the exact details of the models and dataset combinations we sued for experimentation.

#### 7.2.1 RESNET50 ON TINY-IMAGENET

For this training domain, we trained using the Tiny-imagenet dataset CS231N (2015) with resnet50 He et al. (2015). However, we changed the training mechanism in order to get validate our results. Each network we train for 40 epochs, with a batch size of 64. Additionally, we use the Adam optimizer to train with learning rate set to 0.001 and momentum set to 0.9. We also use weight decay set to 0.0001, and we anneal the learning rate to 0.0001 after 30 epochs of training in order to converge faster. We apply the same image transforms as on full Imagenet.

We chose this optimization method because we felt that it achieved a good overall accuracy at a baseline level and represents the results in Sun (2016) in their vanilla model. We do not use the same preprocessing or image transforms in the report Sun (2016). Moreover, we wanted a quick way to estimate how our method would perform on full Imagenet.

#### 7.2.2 RESNET50 ON IMAGENET

Here, we train each network for 90 epochs with a reduced batch size of 128 instead of 256 because 256 would not fit on a GPU in addition to our pruning layers. We found that changing the batch size to 128 but retaining all other hyperparameters as specified in He et al. (2015) we were able to achieve the same benchmark 74.94% accuracy as the paper. We train for 90 epochs with SGD with momentum set to 0.9 and weight decay is $1 \times 10^{-4}$. We set the initial learning rate to be 0.1 and then anneal the learning rate to 0.01 at epoch 30 and then finally to 0.001 at epoch 60.

For dataset transformations, we perform the same transformations as [2]. This means that during training we perform a random sized crop to size 224x224, randomly flip the image horizontally, and normalize the image. The batches are shuffled during training. For validation, we resize the image to 256 and then center crop to size 224x224 and then normalize.

#### 7.2.3 RESNET50V1.5 ON IMAGENET

We train our model for 90/100 epochs and use SGD with momentum (0.9) to optimize. The standard model says that learning rate should 0.1 for 256 batch size, but since that didn't fit in our GPUs with our sparsity mechanism, we used batch size 64 and linearly scaled the learning rate to be 0.05. We set the learning rate decay such that we multiply by 0.1 after 30, 60, and 90 epochs. We have weight decay set to $1 \times 10^{-4}$. Resnet1.5 is defined in detail here [3].

### 7.3 MISCELLANEOUS RESULTS

#### 7.3.1 RESNET50 ON TINY-IMAGENET

Our models actually perform better than the baseline with the following configurations: window pruning with 60% sparsity as well as CK pruning with 20%, 60% and 80% sparsity. The number of

---

[2] https://github.com/pytorch/examples/tree/master/imagenet
[3] https://ngc.nvidia.com/catalog/model-scripts/nvidia:resnet_50_v1_5_for_pytorch

epochs required to reach the converge to the final accuracies is the same for CK and earlier for window at 40% and 60% sparsity.

| Model Sparsity [%] | Window | | | CK | | |
|---|---|---|---|---|---|---|
| | Accuracy [%] | Epoch of Convergence | True sparsity | Accuracy [%] | Epoch of Convergence | True sparsity [%] |
| 0 | 52.03 | 40 | 0.01 | 51.84 | 31 | 0.021 |
| 20 | 50.97 | 40 | 0.48 | 52.40 | 31 | 0.207 |
| 40 | 51.62 | 39 | 0.58 | 51.79 | 31 | 0.404 |
| 60 | 52.09 | 40 | 0.69 | 52.56 | 31 | 0.602 |
| 80 | 51.16 | 36 | 0.84 | 52.07 | 31 | 0.800 |

Table 5: Best accuracy for different sparsity levels for Resnet50 on Tiny-Imagenet with Window and CK Pruning. Both methods see high accuracy close to, or even above the baseline in the cases of Window at 60%, CK at 20%, 60% and 80%. CK converges to the final epoch at the same epoch as the baseline while window converges at the same epoch or earlier in the cases of 40% and 60%.

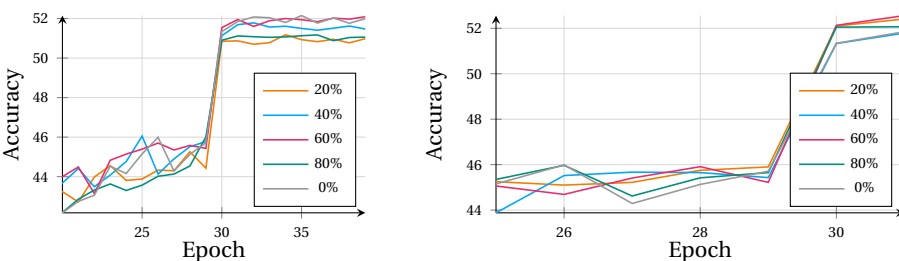

Figure 6: Convergence plots of Resnet50 on Tiny-Imagenet with Window (left) and CK Pruning (right). At all sparsity levels we are the near or above the baseline. Though the differences between models is small, 60% seems to performs the best and is a peak with respect to 40% and 80%.

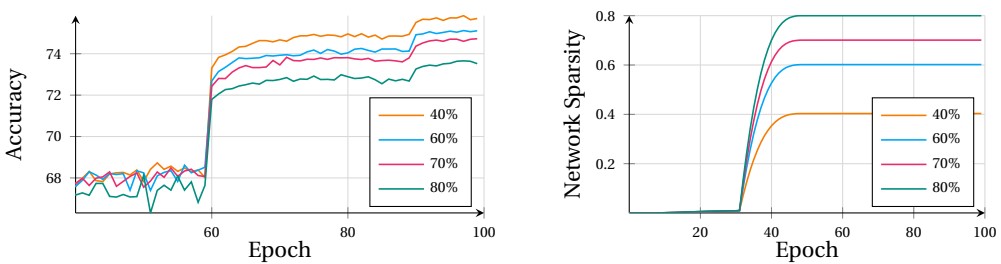

Figure 7: Convergence and sparsity plots of Resnet50v1.5 on Imagenet with CK Pruning, first epoch of sparsity = 30. As the amount of sparsity increases, the accuracy of the model seems to decrease semi-linearly. We can achieve up to 70% while still being above 74% sparsity.

| Model Sparsity (%) | 40 | 60 | 70 | 80 |
|---|---|---|---|---|
| Dense (75.25) | - | - | - | - |
| CK, start 40 | 74.93 (-0.32) | 74.6 (-0.65) | 74.21 (-1.04) | 73.34 (-1.91) |
| CK, start 30 | 74.96 (-0.29) | 74.25 (-1.00) | 73.83 (-1.42) | - |
| CK, start 20 | 74.74 (-0.51) | 73.84 (-1.41) | 73.37 (-1.88) | 72.49 (-2.76) |
| Intra, start 40 | 74.91 (-0.34) | 74.75 (-0.50) | 74.36 (0.89) | 73.22 (-2.03) |
| Intra, start 30 | 74.94 (-0.31) | 74.41 (-0.84) | 73.87 (-1.38) | - |
| Intra, start 20 | 74.73 (-0.52) | 73.65 (-1.60) | - | 72.22 (-3.03) |
| Window, start 0 | - | 72.63 (-2.62) | 71.79 (-3.46) | 69.48 (-5.77) |
| Window, start 30 | - | 74.28 (-0.97) | 73.88 (-1.37) | 72.63 (-2.62) |
| CK, start 40, era 40-70 | - | 74.67 (-0.58) | 74.34 (-0.91) | - |
| Intra, start 40, era 40-70 | - | 74.77 (-0.48) | 74.38 (-0.87) | - |

Table 6: Main results, top-1 accuracy on Resnet50 v1.5 after 90 epochs.

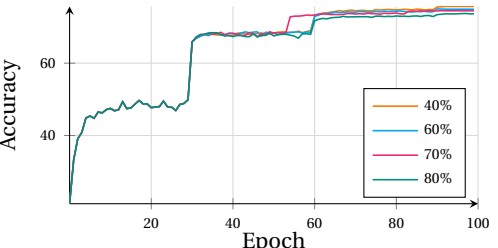 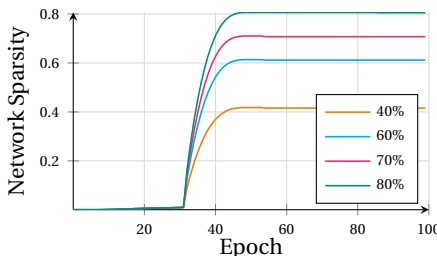

Figure 8: Convergence and sparsity plots of Resnet50v1.5 on Imagenet with Intra-Epoch Pruning, first epoch of sparsity = 30. The left plot shows training over the full 100 epochs instead of zooming in on the tail end of training. This allows us to observe the importance of the learning rate drops at epoch 30 and 60. The drop at epoch 90 does have a small increase, as well.

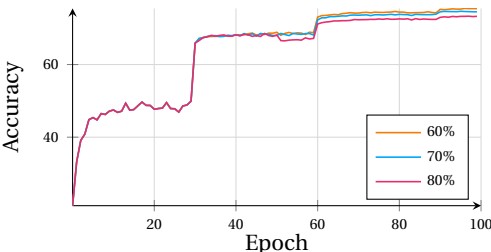 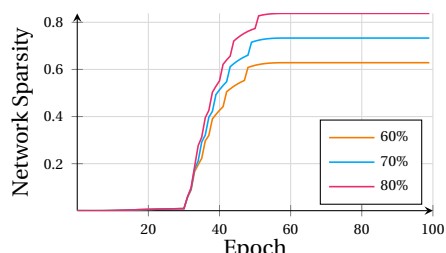

Figure 9: Convergence and sparsity plots of Resnet50v1.5 on Imagenet with Window Pruning, first epoch of sparsity = 30. With window, likw with CK, the impact of the learning rate drops at epochs 30 and 60 is big. Note that the sparsity of the window is not as smooth as CK, showing that it is less uniformly sparse from epoch to epoch.

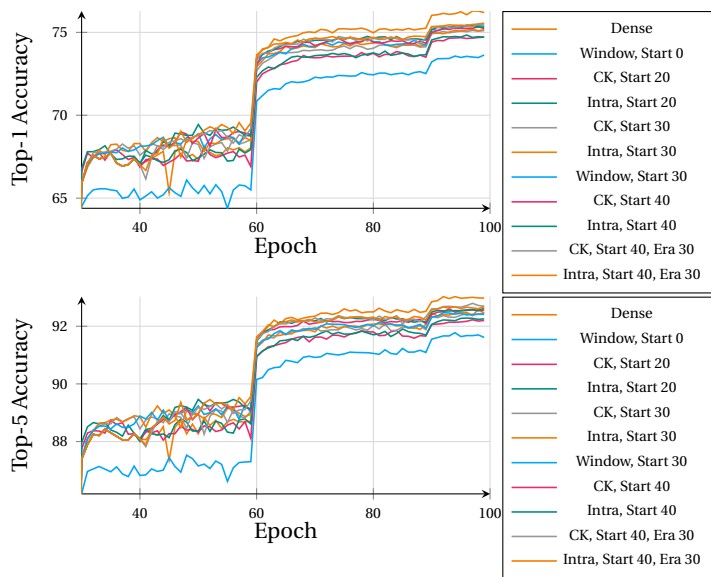

Figure 10: Convergence plots of Resnet50v1.5 on Imagenet at 60%, top-1 and top-5 accuracy. For both top-1 an top-5, window pruning starting at 0 does not perform as well as the other methods. The rest are mostly clustered around 75% top-1 accuracy and 92% top-5 accuracy.

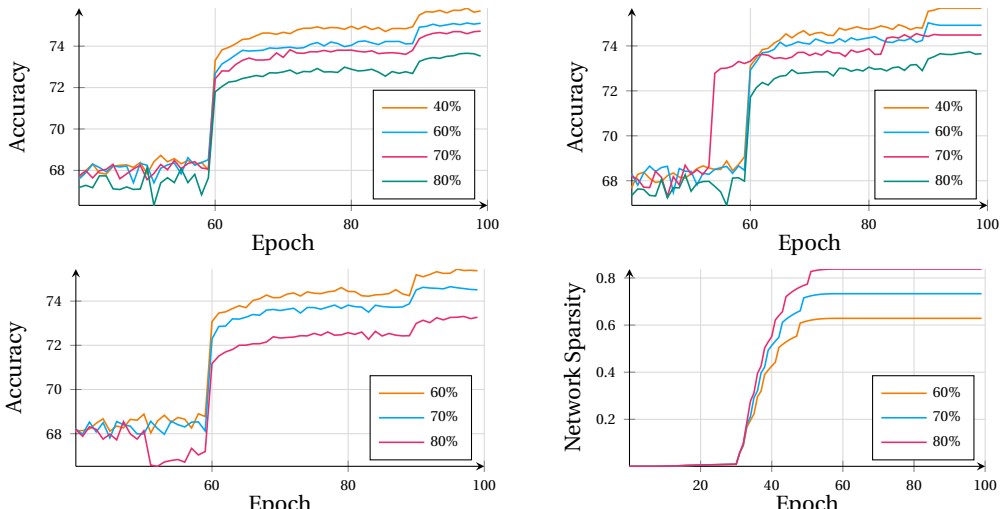

Figure 11: Convergence plots with CK (top left), intra (top right), window (bottom left), and sparsity plot with window (bottom right). Resnet50v1.5 on Imagenet with first epoch of sparsity = 30.

| Model Sparsity (%) | 40 | 60 | 70 | 80 |
|---|---|---|---|---|
| Dense | 1.04 | - | - | - |
| CK, start 40 | 0.89 | 0.73 | 0.71 | 0.83 |
| CK, start 30 | 0.87 | 0.86 | 0.90 | - |
| CK, start 20 | 0.80 | 0.87 | 0.96 | 0.30 |
| Intra, start 40 | 0.98 | 0.64 | 0.72 | 0.79 |
| Intra, start 30 | 0.89 | 0.75 | 0.61 | - |
| Intra, start 20 | 0.99 | 1.10 | - | 0.75 |
| Window, start 0 | - | 1.01 | 1.00 | 0.76 |
| Window, start 30 | - | 1.17 | 0.77 | 0.67 |
| CK, start 40, era 40-70 | - | 0.75 | 0.82 | - |
| Intra, start 40, era 40-70 | - | 0.79 | 0.76 | - |

Table 7: Gain in top-1 accuracy on Resnet50 v1.5 from epoch 90 to epoch 100. This table shows more explicitly that the benefit of the additional 10 epochs of training from epoch 90 to 100 is about 0.8-1.1%.

| Experiment @ 60% Sparsity | Accuracy (Improvement) [%] |
|---|---|
| CK, start 40 | 75.39 (0.05) |
| CK, start 30 | 75.06 (-0.06) |
| CK, start 20 | 74.80 (0.09) |
| Intra, start 40 | 75.46 (0.08) |
| Intra, start 30 | 75.19 (0.03) |
| Intra, start 20 | 74.88 (0.13) |
| Window, start 0 | 73.89 (0.26) |

Table 8: Results when we changed final learning rate drop from epoch 90 to 80, top-1 accuracy on Resnet50 v1.5 at epoch 100. This change does provide 0.1-0.2% improvement is some cases, but the benefit is relatively small.

