# OpenReview forum: "Starfire: Regularization-Free Adversarially-Robust Structured Sparse Training"
_ICLR.cc/2020/Conference — Reject_

### Official Review · AnonReviewer3 · 2019-10-17
**Official Blind Review #3**

**Rating:** 1

**Review:**

This paper introduces a strategy to prune a convolutional neural network during training. To speed up training, the proposed method prunes the weights with the smallest magnitude during only a small number of epochs at the beginning of training, later on continuing training with a fixed sparsity pattern. Several granularity levels for convolutional and fully-connected layers are studied. Furthermore, the robustness of the resulting pruned networks to adversarial attacks is investigated.

Originality:
- As acknowledged at the beginning of section two, the general pruning strategy used here is very similar to that introduced by Narang et al., 2017. While the authors argued that the threshold is computed in a different manner, it also increases gradually during training, as in Narang et al., 2017.
- I acknowledge that Narang et al., 2017 focuses on RNNs, while here the focus is on CNNs. However, the originality of the different pruning strategies used here for convolutional and fully-connected layers is very limited. In essence, these strategies directly follow those studied by Mao et al., 2017.
- The study of robustness to adversarial attacks, while interesting, is also not novel per se, as the idea of performing such a study was proposed in Wang et al.,  2018. I acknowledge that the conclusions drawn here differ from those in Wang et al., 2018. However, there are no explanations for this different behavior.

Methodology:
- While the beginning of Section 2 states that the pruning threshold gradually increases during training, the specific way this is achieved is not clearly explained.
- The pruning strategies depicted by Fig. 2, whether for convolutional layers or for fully-connected ones, never aim to remove entire output channels. However, the only way to truly end up with a smaller network is to remove entire channels and/or layers, as argued in Wen et al., 2016 and in Alvarez & Salzmann, NIPS 2016, as well as studied in Mao et al., 2017 via the filter-level granularity. It is unclear to me how speed would be affected by having a network with the same number of channels and layers, but many parameters set to zero.

Experiments:
- The experiments show the good behavior of the proposed algorithm in terms of sparsity vs accuracy tradeoff. However, while the introduction seems to focus on the benefits of the proposed method in terms of training speed, these benefits are not demonstrated in the experiments, where no timings (neither for training not for inference) are reported.
- As mentioned above, it is not clear to me that the speedup will be significant if the sparsity pattern does not remove entire channels, but I am willing to be proven wrong.

Summary:
My main concern about this paper is its novelty, as the method essentially uses the method of Narang et al., 2017, albeit with a different threshold, with the sparsity patterns of Mao et al., 2017. The experiments demonstrate that the method is effective at pruning, but do not provide any timings to evaluate the resulting speedups.

**Experience Assessment:**

I have published one or two papers in this area.

**Review Assessment: Checking Correctness Of Derivations And Theory:**

I carefully checked the derivations and theory.

**Review Assessment: Checking Correctness Of Experiments:**

I assessed the sensibility of the experiments.

**Review Assessment: Thoroughness In Paper Reading:**

I read the paper thoroughly.

---

> ### Author Response · Authors · 2019-11-14
> **Author Response**
>
> (1) We agree that our method is similar to Narang et al.; however, achieving high levels of accuracy and high levels of structured sparsity for CNNs was missing; we contribute substantial experiments to find the limits of the final level of sparsity and how few epochs we can train for CNN. Also, Narang et al. prune after every iteration while we prune after every epoch (for as little as 20 epochs out of 90), so there are much fewer updates in our method. We grant that the adversarial is not exactly novel. However, our goal was to investigate the robustness in presence of structured sparsity.
>
> (2) Thank you for pointing out that this is unclear. The gradual pruning is done as shown in Fig 1, but we will add prose to explain the process. Our goal is to provide structured sparsity, rather than speed up training, that is we would like to reduce the memory footprint but also structure it in a highly regular way that could be exploited by a hardware accelerator. As you say, this is setting parameters to 0, which is the goal of many. Please see Table 4 for a number of them. On top of that, all the hardware accelerators put mechanisms to skip computations with zeros please see reference [Han 2015b] for an example.
>
> (3) The background in the introduction regarding speedup seems to have mischaracterized our main intentions, so we will clarify that it is mainly to provide structured sparsity while maintaining accuracy. However, we would like to note that we noted we would get speedups from sparsification because we would reduce the total number of computations [Zhu et al 2019] [Parashar et al 2017] by having a small number of epochs in which we do pruning and fixing the sparsity masks to their final values early in training rather than sparsifying until the end of training or after training.
>
> Parashar, Angshuman, et al. "SCNN: An accelerator for compressed-sparse convolutional neural networks." 2017 ACM/IEEE 44th Annual International Symposium on Computer Architecture (ISCA). IEEE, 2017.
>
> Zhu, Maohua, et al. "Sparse Tensor Core: Algorithm and Hardware Co-Design for Vector-wise Sparse Neural Networks on Modern GPUs." Proceedings of the 52nd Annual IEEE/ACM International Symposium on Microarchitecture. ACM, 2019.

---

### Official Review · AnonReviewer2 · 2019-10-26
**Official Blind Review #2**

**Rating:** 1

**Review:**

SUMMARY
-------

This paper explores a series of incremental variations of existing pruning techniques for compressing Resnet-50 for ImageNet. Specifically, it proposes concentrating all pruning during an early "era" of training (the first 20-50 epochs out of 100 total). It also explores hybrids between sparse pruning and structured pruning. Finally, it considers the adversarial robustness of the resulting networks to the FGSM attack.

This paper makes no novel proposals and experiments are minimal. There are no clear takeaways from the results of these experiments. The goals of the paper are unclear, and it is difficult to compare this paper to existing work.

This paper has no clear motivation and makes no tangible contributions to the literature and, therefore, I recommend a rejection.

CONTRIBUTIONS
-------------

1) A study of the appropriate window ("pruning era") for pruning Resnet-50 on ImageNet and TinyImageNet
2) A study of the tradeoffs between various forms of structured and unstructured pruning.
3) An analysis of the adversarial robustness of the pruned networks.


DETAILED COMMENTS
------

PROBLEMS ADDRESSED

It was challenging to discern the specific problems that this paper sought to address and, relatedly, the goals that the paper sought to achieve. The introduction of the paper lists a wide variety of problems in the existing literature:

1) Paragraph 3: Structured sparsity introduces "regularization and computational overhead."
2) Paragraph 3: "Coarse-grained sparsity" cannot eliminate enough parameters to perform well on "edge devices."
3) Paragraph 4: Dynamic sparsity techniques require more training epochs.
4) Paragraph 4: Dynamic sparsity techniques do not preserve network accuracy (1-2 percentage point drop at 80% sparsity).
5) Paragraph 4: Dynamic sparsity requires reconfiguring the sparsity pattern frequently, which is computationally expensive.

The paper does not justify the fact that any of these are actually problems, nor does it make any attempt to quantify the extent of these problems. Moreover, the proposed techniques do not resolve any of these problems. Corresponding to the numbers above:

1) The paper never measures this overhead nor justifies that it is a problem in practice. Meanwhile, the techniques proposed in the paper introduce substantial overhad of their own, including training for an extra ten epochs. It is possible that the techniques proposed in this paper have worse overhead than the techniques that are criticized in the introduction. Since the paper provides on numbers either way, it is impossible to tell. In short, computational cost is a key part of the author's argument despite the fact that there is no empirical support for any of these claims.

2) I believe the paper means that, in order to get to sufficient levels of sparsity to work on "edge devices," accuracy drops unacceptably far. What does the paper mean by "edge devices," what are sufficient levels of sparsity, and what does it mean for accuracy to drop unacceptably far? The paper has numbers for the proposed methods, so it should be possible to make this comparison if such baselines are explicit.

3) The proposed techniques also require the same number of additional training epochs, so this complaint is unaddressed.

4) The proposed techniques show a 2-3 percentage point drop at 80% sparsity (Table 1), which is actually worse than the technique that the authors criticize.

5) The proposed techniques require pruning after every single training step during the "pruning era." This is likely to be more computationally expensive than any of the other gradual pruning and dynamic sparsity techniques listed, which prune at intervals of hundreds or thousands of iterations. In addition, the authors never justify why changing the sparsity pattern frequently throughout training will affect performance. On GPUs with modern frameworks, I see no reason why this should matter so long as the sparsity pattern does not change too frequently (although that is exactly what this paper proposes to do during the "pruning era").


GOALS

It was also challenging to discern the goals of the paper. Was it:

1) To produce the smallest possible trained networks with the highest possible accuracy?

2) To reduce the cost of obtaining a pruned network for inference-time? (Or to reduce the cost of obtaining a sufficiently efficient pruned network for inference-time?)

3) To reduce the cost of training neural networks in general by pruning them during training?

In the introduction and the related work section, these goals go unstated, making it difficult to determine how this paper compares to existing work. The comparisons provided in the paper focus on specific aspects of each related work rather than the entire picture. For example, in comparison to Mao et al., the authors claim better accuracy at one sparsity level, implying goal 1. However, for to Lym et al., the paper focuses on the computational costs of training the network, implying goal 3.


UNJUSTIFIED CLAIMS ABOUT NEURAL NETWORK COMPUTATION

Throughout the paper, there are a number of unjustified claims about which neural network configurations will perform better on contemporary hardware. Considering computational efficiency appears to be a key element of the paper's argument, these claims require citations or - particularly when various configurations are compared to one another - empirical support. Some examples:

* Section 1, Paragraph 3: "The regularization term [of structured sparsity] modifies the original training and can be expensive in hardware."
* Section 1, Paragraph 3: "The final network [from Lym et al. 2019] contains an insufficient degree of sparsity for deployment on edge devices."
* Section 1, Paragraph 4: "Continuous reconfiguration of the sparsity pattern is expensive as it does not allow for compression of weights during training"
* Section 1, Paragraph 5: "having a fixed sparse multiply-accumulate pattern allows weight compression during training and can save compute and energy in hardware"
* Section 5, Paragraph 2: "A strict parameter allows the hardware mapping to plan for a fixed number of multiply-accumulate operations."
* Section 5, Paragraph 2: "Regularization, although useful in forcing the network to learn prunable weights, adds more irregularity to computation flow."


PRUNING TECHNIQUES

* Recomputing the pruning mask at every training step seems gratuitously inefficient.
* Sorting the weights in the entire network shouldn't be particularly inefficient if it isn't done on every single iteration. (Section 2.1 paragraph 1)
* Why do you maintain the same number of weights in each convolutional filter with window pruning? (Presumably for performance reasons, but you never say that.)
* None of the pruning methods are novel. They're simply various permutations of structured and unstructured magnitude pruning as proposed by many others in the literature.


EXPERIMENTS

* Section 3.1 Paragraph 2: It appears that you are exploring the best "pruning era." If you are to do so, you will have to sweep over (1) the length of the pruning era (2) the starting epoch of the pruning era, and (3) the shape of the function used to determine sparsity. Instead, it sounds like you tried two arbitrary pruning eras (0-30 and 30-50). Likewise, in Paragraph 3, you test only a small number of possible scenarios.
* Section 3.1 is generally hard to parse. It is unclear what you are studying. The ideal pruning era? The relative performance of the pruning methods introduced in section 2?
* How many times did you replicate each experiment? You should ideally include at least 3 (and preferrably 5) replicates with mean and stddev reported.
* What baselines are you including? You should include a random pruning baseline and you should ideally replicate any methods that you compare to.


RESULTS

* Section 4.1: The data you refer to is in an appendix even though it is crucial to the main body of the paper. The appendices should contain material that is nonessential for making sense of the paper.
* Section 4.2 Paragraph 1: Are these numbers good? A standard sparse pruning technique (Gale et al. 2019, https://arxiv.org/pdf/1902.09574.pdf) achieve 70% sparsity without any change in accuracy. Please include baselines comparing to other methods in the literature.
* Table 2: It is difficult to compare the results in these papers. PruneTrain aims to reduce the cost of training and measures cost reductions in FLOPS. If you intend to compare against this paper, you should quantify the cost of training using your method against that of PruneTrain. Merely presenting sparsity and accuracy numbers is insufficient. Likewise for the dynamic sparsity experiments. What is your goal in showing this comparison, and did Mostafa and Wang share that goal when they justified their technique?
* You do not describe the hyperparameters for Intraepoch pruning (the balance between window and CK - last paragraph of 2.1.1)


ADVERSARIAL ROBUSTNESS

Considering the fact that this paper focuses on proposing new variations of existing pruning techniques, any discussion of adversarial robustness seems to be (1) out of place and (2) an afterthought. If the authors delete a half-page of content (one phrase from the abstract, a paragraph and bullet from the introduction, and a paragraph each from sections 3 and 4), this content could be removed with minimal impact to the paper's main contributions. The content on adversarial robustness is cursory, uses a weak and out-of-date attack (FGSM), and does not compare to any other pruning methods. In fact, the one comparison is to the results in a paper (Wang et al, 2018) that looks at both FGSM and PGD (a stronger attack) on completely different networks and tasks (MNIST and CIFAR10). The paper would be stronger if content on adversarial robustness was removed entirely.


OTHER MINOR COMMENTS

* The title includes the word "starfire," but it never appears again in the paper. The paper proposes no specific technique, so there isn't anything to name.
* Use the \ begin{appendix} command before you create the appendices and the \ end{appendix} command when you are done. You can then use \section normally and each section so-created will appear with a letter rather than a number.
* Figure 4 is very hard to read.

**Experience Assessment:**

I have published one or two papers in this area.

**Review Assessment: Checking Correctness Of Derivations And Theory:**

N/A

**Review Assessment: Checking Correctness Of Experiments:**

I assessed the sensibility of the experiments.

**Review Assessment: Thoroughness In Paper Reading:**

I read the paper thoroughly.

---

> ### Author Response · Authors · 2019-11-14
> **Author Response Part 1**
>
> (1) This paper studies the sparse training for maximum convergence accuracy of Resnet50  with Imagenet and delivers the best result for sparse training with static mask for over 70-80 epochs of training schedule in terms accuracy and acceleration potential.
> The experiments performed all the sensitivity studies and report over 34 experiments covering various levels of sparsity, length pruning era, start epoch of pruning era, both Resnet50 v1 and v1.5, and various sparsity granularities just in Table 1 alone.
> To give a context some of the main references in Table 2 report less than 10 data points.
>
> (2) Thanks for you for comment regarding the problems we highlighted with current literature. We will cite literature to address all the above.
> Just to give context:
> 1) Regularization contains operations such as vector norm (normalization, divide, square root) that are not typically multiply accumulate nature and hence are expensive for accelerators and GPUs as they are atypical with high latency.
>
> 2) Sorry for the miscommunication, our meaning with this sentence was that the levels of sparsity in previous coarse grain sparsity methods were not high-enough for a given convergence accuracy. Therefore, the overheads incurred to handle sparsity (indexing/compression/decompression) were not justified. Our main point is that edge devices are one example area where a high level of coarse grained sparsity while maintaining accuracy is desirable, and this paper achieved this goal. This paper also achieves similar goals for sparse training that we will address below.
>
> 3-4) This is supported by Table 2: given a target convergence accuracy Mostafa and Wang had to train for an extra 10 epochs (100 epochs) to be able to reach within 1% of a 90 epoch baseline. We showed our results at all relevant epochs and the difference to baseline both at 90 and 100 epochs.
>
> 5) This reconfiguration happens throughout the whole network. Restructuring the sparsity masks is extremely expensive in any system. It includes decompression and recompression which in turn triples memory accesses.
> Replacing the non-zeros and refreshing the indexes is a memory intensive operation. It has high latencies and high energy consumption. However ours only happens after every epoch. This was represented as after every step in Algorithm 1, and we will correctly update the figure to accurately represent our mask update schedule.
>
>
> (3) We can show the nature of the computation incurred for reparametrization. While the point is taken that we have no empirical study to support this, we make the argument that our overheads are minimal given that they are only incurred during 20% of total training schedule and that we eliminated regularization overheads. Also, we update the sparsity only for a total of 20 times once/epoch during pruning era (again this was misrepresented in Algorithm 1 and will be corrected).
>
> (4) The reason we mentioned edge devices was to highlight that the accuracy drop does not justify using low levels of sparsity because the overheads of indexing/compression/decompression are higher than just keeping data dense. [https://en.wikipedia.org/wiki/Sparse_matrix] [https://kkourt.io/phd/phd-en.pdf], [https://arxiv.org/pdf/1901.07827.pdf].
> For example, CUSparse library performs only better than CUBLAS when degrees of sparsity are over 90% (https://developer.nvidia.com/cusparse)
>
> (5) We show both convergence at epoch 90 and 100 for comparison to baseline and all other methods. The argument here is that those extra 10 epochs with high degrees of sparse training are much cheaper. The epoch 90 results are in Table 6 in the appendix.
> Even compared to 90 epoch baseline we drop less than 1% in accuracy.
>
> (6) Dynamic sparse reparametrization is impossible to implement on any target architecture because it changes sparsity mask every step. This means changing indexes of non-zeros and decompression/recompression at every step. This is basically infeasible on any accelerator. We contacted the authors and they told us they trained the network like a dense network. Our goals are clear: we want fixed sparsity mask after certain epoch and we wanted to eliminate any irregular computation that is expensive on accelerators.
>
> (7) This a mistake on our behalf in the algorithm text and it is actually after each epoch in our code. Thank you for pointing this out. We only prune total of 20 times (length of pruning era).

---

> > ### Author Response · Authors · 2019-11-14
> > **Author Response Part 2**
> >
> > (8) Our main goal is to make training sparse coarse and simple so an accelerator can exploit savings in sparse computation. So, yes, our goal is to produce the smallest possible trained networks with highest possible accuracy but with a fixed sparsity after a certain epoch and minimum irregular computations to easily deploy on hardware accelerators. We also want to both reduce the cost and also provide a final network that is sparse enough to be deployed without further pruning. We do not focus on reducing the cost of obtaining a pruned network for inference-time. We will make these goals clearer in the introduction.
> >
> > (9)
> > RESPONSES TO UNJUSTIFIED CLAIMS (in order)
> >
> > The following reference shows the regularization term in the loss of the neural network. This computation requires irregular computation such as norms or square roots that are expensive in hardware. [https://www.analyticsvidhya.com/blog/2018/04/fundamentals-deep-learning-regularization-techniques/]
> > [http://faculty.washington.edu/nrsimon/standGL.pdf]
> >
> > -This is because the overheads incurred for compressing/indexing would make it slower than when run in dense format, so these overheads need to be offset higher sparsity levels.[https://kkourt.io/phd/phd-en.pdf], [https://arxiv.org/pdf/1901.07827.pdf]
> >
> > -This only requires Index changes. Weights can still be compressed.
> >
> > -After a certain point in training there won’t be any recompression and the indirect access to indexes due to compression will be fixed and can be cached. This way the compressed weight matrix can be stored without modifying the compression infrastructure. Only the value of data is changed instead of its location to avoid complicated compression/recompression as in other solution.
> >
> > -We showed that for window pruning we can always use only 4 MAC units hence using half computation.
> >
> > -As discussed above, the regularization often requires the vector norm which requires division (to avoid overflowing) and square root that are very expensive in hardware.
> >
> > -This should be epoch, again our mistake.
> >
> > -This should be epoch, again our mistake.
> >
> > -This answers your question for Section 5 for performance on a fixed number of FPUs. We will make this more clear in the paper.
> >
> > -Our goal is characterization while maintaining accuracy, reducing sparsity masks, and eliminate irregular computations.
> >
> > We performed experiments with 0-30, 20-40, 30-50, 40-60 in Table 1 and Table 6 (first column).
> >
> > -Both, we will clarify.
> >
> > -You are right but these 34+ experiment datapoints took over 4 months and so were intractable to replicate 3-5 times. We will perform that on select datapoints.
> >
> > -Random pruning is not actually studied in any of the previous work. We cannot replicate every method we compared as we trust the previous work and the validity of their results.
> >
> > -We believe appendix data is providing additional evidence for our findings. We put the main findings in the limited 8 pages of the paper.
> >
> > -While the loss in accuracy is higher, we provide structured sparsity and fix the structure early on in training, meaning unlike in magnitude-based pruning, we do move the weights around after the structure is fixed. This can simply not be accelerated by hardware. We will add this to our citations and our comparisons in Table 4..
> >
> > -Prunetrain savings is not justified given their loss in accuracy and their low degree of sparsity. One can simply train a dense network for less number of epochs and stop at their achieved accuracy and save more computations without the hassle of training and restructuring the network.
> > Dynamic sparse is infeasible for any target architecture to gain performance.
> > Please note that these are the closest sparse training papers focused on CNNs.
> > Dynamically reconfiguring sparse masks is not feasible for accelerators as it needs decompression/recompression at each step.
> > Mostafa and Wang studied the upper limits of sparsity while training and maintaining accuracy.
> > We want high levels of accuracy but we want to eliminate decompression/recompression at each step. Unlike prunetrain our accuracy levels are much higher and the network mask is also fixed.
> >
> >
> >
> > (10)
> > * You do not describe the hyperparameters for Intra-epoch pruning (the balance between window and CK - last paragraph of 2.1.1)
> > -Thanks we will add that
> >
> > (11)
> > ADVERSARIAL ROBUSTNESS
> >
> > -Thank you for the suggestion, we will look into adding PGD attacks or consider completely removing this aspect.
> >
> > (12) Thank you for your other comments regarding the name, appendix, and figure clarity. We will fix these.

---

### Official Review · AnonReviewer1 · 2019-10-26
**Official Blind Review #1**

**Rating:** 1

**Review:**

The paper investigates methods to train neural networks so the final network has sparse weights, both in convolutional layers and in fully connected layers. In particular, the paper focuses on modifying the training so that the network is first trained without sparsification for a certain number of epochs, then trained to be increasingly sparse, and then fine-tuned with a fixed sparsity pattern at the end.

While I find the overall approach of the paper interesting, currently the experiments are not systematic enough to derive clear insights from the paper. Hence I unfortunately recommend rejecting the paper at this point. I hope the authors find time to conduct more systematic experiments for a future version of the paper.

Concretely, the following would be interesting experiments / questions:

- How effective is the proposed training method on architectures other than ResNets?

- What happens if the "pruning era" is made longer, started substantially earlier, or started substantially later? Currently it is not clear if the epoch 30 - 50 pruning era is (approximately) optimal and how much performance varies with begin and end of the pruning era.

- Due to the small variation between some of the methods, it would be good to investigate how robust the ordering is when the experiment is re-ran with different random seeds etc.


In addition, I have the following suggestions:

- The authors may want to remove or enhance the adversarial robustness evaluation. Currently the authors only evaluate robustness against FGSM, but it is well known that iterative attacks such as PGD are more effective.

- Instead of "intra-epoch pruning" or "intra", the name "combined" may be more clear for the combined method.

- In the description of the experimental setup, it could be good to specify what GPUs were used (since this lead to the smaller batch size).

- It could be helpful for the reader to discuss how predictive results on Tiny-ImageNet are for results on ImageNet.

- In Table 2, it would be good to add context by comparing to prior work with sparsity level 60% and some of the compression-focused methods from Table 4.

- In the comparison to Mao et al. (2017), it could be good to clarify that they also work with ResNet models on ImageNet.


**Experience Assessment:**

I do not know much about this area.

**Review Assessment: Checking Correctness Of Derivations And Theory:**

N/A

**Review Assessment: Checking Correctness Of Experiments:**

I assessed the sensibility of the experiments.

**Review Assessment: Thoroughness In Paper Reading:**

I made a quick assessment of this paper.

---

> ### Author Response · Authors · 2019-11-14
> **Author Response**
>
> (1) Our argument for only using one architecture is that Resnet networks are the benchmark for MLPerf and the hardest/longest networks to train. All other recent work on CNNs use Resnet, on Imagenet and some smaller datasets, as well. However, to address this concern we will add experiments on VGG.
>
> (2) The main goal of this approach is to restrict the pruning era to reduce complexity especially on accelerators. We also aimed to reach a fixed sparsity mask as early as possible. Keeping the above goal in mind, we did perform sensitivity analysis (shown in Table 1) keeping the final convergence accuracy as high as possible. We demonstrated that shrinking the pruning era damages the accuracy. On top of that we moved the pruning era 10 epochs earlier or later in training and studied wider pruning schedules.
>
> (3) This is a good suggestion; however, what prevented us from doing this is that these experiments we demonstrated each take several days of training per data point. To just perform the sensitivity analysis on the width of the pruning era it took us several months. We can definitely perform these analysis for smaller networks and datasets.
>
> (4) Thanks for the suggestion regarding adversarial attacks. We will investigate PGD. However, we believe that finding that structured sparse training is robust for FGSM is still valuable as other work show.
>
> (5) Thanks we will fix the name of intra-epoch pruning to combined method to improve clarity.
>
> (6) We used RTX2080 instances. We will add that to the paper.
>
> (7) Very valid point regarding the productivity of the Tiny-Imagenet results. We will add this.
>
> (8) Regarding Table2, compression focused methods take around 180 epochs of training if aiming for levels of accuracy that reported. If not, they have much worse accuracy numbers without providing structured sparsity and without the potential of computation savings during training. So, we chose to only compare with sparse training methods.
>
> (9) Thanks, we will clarify that Mao et al. (2017) worked with Resnet/Imagenet.

---

### Decision · Program_Chairs · 2019-12-19

**Decision:**

Reject

**Comment:**

This paper concerns a training procedure for neural networks which results in sparse connectivity in the final resulting network, consisting of an "early era" of training in which pruning takes place, followed by fixed connectivity training thereafter, and a study of tradeoffs inherent in various approaches to structured and unstructured pruning, and an investigation of adversarial robustness of pruned networks.

While some reviewers found the general approach interesting, all reviewers were critical of the lack of novelty, clarity and empirical rigour. R2 in particular raised concerns about the motivation, evaluation of computational savings (that FLOPS should be measured directly), and felt that the discussion of adversarial robustness was out of place and "an afterthought".

Reviewers were unconvinced by rebuttals, and no attempts were made at improving the paper (additional experiments were promised, but not delivered). I therefore recommend rejection.